# Hepatitis B Virus-Associated Hepatocellular Carcinoma and Chronic Stress

**DOI:** 10.3390/ijms23073917

**Published:** 2022-04-01

**Authors:** Nicholas Noverati, Rukaiya Bashir-Hamidu, Dina Halegoua-DeMarzio, Hie-Won Hann

**Affiliations:** 1Department of Medicine, Thomas Jefferson University Hospital, Philadelphia, PA 19107, USA; nicholas.noverati@jefferson.edu (N.N.); rukaiya.bashir-hamidu@jefferson.edu (R.B.-H.); dina.halegoua-demarzio@jefferson.edu (D.H.-D.); 2Division of Gastroenterology and Hepatology, Thomas Jefferson University Hospital, Philadelphia, PA 19107, USA

**Keywords:** Hepatitis B virus, hepatocellular carcinoma, chronic stress, hepatocarcinogenesis

## Abstract

The Hepatitis B virus is one of the most significant hepatocarcinogens globally. The carcinogenic mechanisms of this virus are complex, and may include interactions with the host’s immune system. Certain factors, such as stress on the body, can also potentiate these mechanisms. Stress, although adaptive in an acute form, is deleterious to health when chronic and can both suppress and activate the host’s defense system. In hepatocellular carcinoma, this can lead to tumor initiation and progression. Those that are more prone to stress, or exposed to situations that incite stress, may be at higher risk of developing cancer. Racial disparities, for example, are a source of chronic psychosocial stress in America and predispose minorities to poorer outcomes. As it remains perplexing why some individuals with chronic hepatitis B develop feared complications while others do not, it is important to recognize as many risk factors as possible, including those often overlooked such as chronic stress.

## 1. Introduction

Primary liver cancer, of which hepatocellular carcinoma (HCC) is the most common, was the sixth most diagnosed cancer globally in 2020 [1]. HCC is thought to arise from ongoing stressors to the liver, one of which can be a chronic hepatitis, such as that from the Hepatitis B virus (HBV).

The Hepatitis B virus is still very much globally present and an important risk factor in the development of HCC [2]. Since its discovery in 1965 and first vaccine in 1983, there has been a decline of infections during the past decades [3,4]. Indeed, vaccinating against Hepatitis B along with treatment advances for Hepatitis B and C have been shown to shift risk factors for primary liver cancer towards that of obesity and diabetes [4,5,6]. However, the World Health Organization (WHO) still estimated in 2019 that 296 million people were living with chronic hepatitis B infection and 820,000 deaths were from HCC or cirrhosis as complications of HBV infection [7]. Thus, it is relevant to understand more about HBV-associated HCC and which individuals with chronic HBV infections progress to this grim diagnosis.

It is unclear why some people with HBV infection that are mostly asymptomatic for years develop HCC, while others do not. Even more perplexing is when this phenomenon is observed within families. For example, significantly divergent long-term outcomes have been observed among siblings with vertical transmission of presumably the same HBV viral genotype [8,9,10]. This shifts the focus from virus to host. What within the host leads to these vagaries? One proposed theory is the effect that chronic stress can pose as a risk factor for developing HCC, in particular.

It is important to define what stress is and how it interacts with the body. Stress is a vague and rarely distinctly defined term [11]. The concept of “stress” was first conceived by scientist Hans Selye, who thought of it as anything that disrupted the process of homeostasis, or maintaining balance within the body [11,12,13]. Since its first conception, stress has not only been further defined, but understood to elicit certain responses within the body in attempt to maintain a degree of homeostasis. We now understand a complex interaction between the nervous system and hormone release, such as the hypothalamus-pituitary–adrenal axis. Further, we define stress and stress responses temporally as acute or chronic, and by qualifiers such as psychological, physical, and/or social. Although stress is often received in a negative connotation, an acute form is thought to be generally more adaptive and beneficial. Conversely, in a chronic form, it can be deleterious to one’s health. Selye himself, the “father of stress,” even hypothesized that chronic stress can lead to bodily dysfunction [11,13,14].

What effects might chronic stress have on the body? Although not fully understood, some have theorized about the different effects chronic stress can impose. There has been specific interest in the literature over the years on how chronic stress can cause imbalance within the immune system response and lead to increased morbidity and mortality from a multitude of disease [14,15]. Of note, cancer incidence and chronic stress have also been shown to correlate [16].

The focus on this article is to better understand, through review and synthesis of the literature, the role that chronic stress plays as a risk factor in developing HBV-associated HCC. This will be done by reviewing and synthesizing the basics of the Hepatitis B virus, theorized mechanisms of HBV-associated HCC, the interplay between chronic stress, the immune system and liver disease in general, and existing literature on HBV-associated HCC and chronic stress. We will also briefly review psychosocial chronic stress and racial disparities as a possible risk factor for developing HBV-associated HCC.

### Hepatitis B Virus

Hepatitis B virus (HBV) was first discovered in 1965 by Baruch Blumberg. He identified a new antigen in the sera of transfused patients with hemophilia, later finding it to be the hepatitis B surface antigen (HBsAg) [3,17]. By 1980, the entire genome of the virus was sequenced with the aid of advances in technology. Soon after, a vaccine became available in 1983 [17]. However, the virus continues to be endemic in many regions of the world, including Southeast Asia, China, and a majority of Africa [18]. These areas are still developing and do not have access to as many resources such as vaccines. In sum, there are still nearly 300 million cases of HBV globally [7]. Our knowledge surrounding this virus, how it replicates, and contributes to pathology in the human body has grown tremendously since these early beginnings.

HBV is a DNA virus within the Hepadnavirus family that has a unique replication process and pathogenesis. Within a cell, virus DNA is converted into RNA before reverse transcription back into DNA. As the virus continues this process in its replication, the immune system begins to respond, causing injury to the liver. Indeed, the virus itself is not actually cytopathic. Further, the phases of immune response can be labeled based on the activity of viral replication (measured by HBV DNA, hepatitis B e-antigen (HBeAg)) and the response from the host (ALT levels, anti-HBe and anti-HBs) [18].

The first phase of immune system response is termed immune-tolerant, when ALT levels reflecting injury are still low, while DNA levels are high. As the immune system starts to “fight” back in the immune reactive phase, it causes elevations in ALT as HBV-infected hepatocytes are destroyed. Eventually, this leads to the inactive carrier phase with lower levels of HBV DNA and ALT, while HBsAg persists. Resolution of infection is marked by disappearance of HBsAg [18].

The goal of treatment of HBV is to decrease viral replication to prevent ongoing liver injury that could lead to hepatic decompensation and the development of HCC. Elevated serum HBV DNA level (10,000 copies/mL) is a strong risk predictor of hepatocellular carcinoma independent of HBeAg, serum alanine aminotransferase level, and liver cirrhosis [19]. Treatment options include interferon, which interferes with the immune system response to the virus and has potential anti-viral effects [17,20]. Other medications that interrupt viral replication by inhibition of reverse transcription include nucleoside analogues (NA) such as lamivudine, adefovir, telbivudine, entecavir, tenofovir disoproxil fumarate, and tenofovir alafenamide [17,18]. Reduction of incidence of HCC by NAs has been well documented [21,22,23,24]. On the other hand, persistent risk for development of HCC despite successful suppression of the virus has also been reported [25,26,27,28,29,30]. Most recently, poorer survival was observed in patients who developed HCC despite long-term successful viral suppression, when compared to those who developed HCC with no prior antiviral therapy [31]. All these reports enforce the need for HBV cure drugs. Several potential HBV cure drugs are currently in development and in early phase trials [31,32,33,34]. These offer the hope of eradicating HBV and are discussed further below.

## 2. HBV-Associated HCC: Proposed Mechanisms and Treatment Options

There are two broad themes of proposed HBV-associated HCC pathogenesis: those related to viral–host integration and those of virus–immune system interaction.

As HBV replicates, it has been shown to incorporate into host DNA during the acute phase of infection, leading to oncogenesis through different mechanisms [30]. For example, studies have shown portions of HBV DNA acting directly and indirectly to activate genes and form oncogenic proteins [2,35]. Insertions also have been shown to cause duplications, deletions, and translocations within chromosomes [2,35]. These hepatocarcinogenic mechanisms are further perpetuated by interactions with the host immune system.

Chronic HBV infection can lead to periods of immune activation in response to the virus, which injure the liver and increase the risk of tumorigenesis. These events are usually evident through the presence of elevated ALT. Immune cells that respond to the virus have many downstream effects: they release cytokines and chemokines that can contribute to cancer growth, injure liver cells that then causes rapid proliferation and increased risk of mutagenesis, and contribute to reactive oxygen species production that cause further DNA damage [2]. As explained by Ringelhan et al., this process can also be understood by considering the effect that persistent HBV infection has on dysregulating the liver’s baseline immune tolerant state [36]. Since the liver filters many potential pathogens as it receives an abundance of blood flow from the gut, there is a degree of tolerance from the immune cells within to prevent recurrent immune-mediated damage. However, when infection from a virus like HBV is present, proinflammatory cytokines are upregulated and break this tolerance, allowing the immune system to be activated and incite tumorigenesis [36]. Once tumor is formed, further research into the interaction with the immune system has shown that cells such as T regulatory cells and cytokine profiles switch towards that of preserving tolerance and allowing HCC to progress [37,38]. Because of the carcinogenic potential of HBV, treatment of the virus is paramount. Better understanding of HBV–HCC pathogenesis has also lead to advancements in treatments of both the viral infection itself and cancer, if present.

Current antiviral treatment of HBV has reduced the risk of developing HCC in the past [21,22,23,24]. Not only does treating HBV help prevent HCC, concurrently treating it at the time of HCC diagnosis and alongside tumor-specific therapies has been shown to improve survival [30,39,40,41,42,43,44,45,46]. However, persistent risk of developing HCC despite “successful” treatment has been observed [25,26,27,28,29,30]. For example, this was observed retrospectively in a cohort of patients at Thomas Jefferson University Hospital who had successful viral suppression for over 10 years [27,28]. This is because treatment only suppresses the virus from replicating but does not eradicate it. Viral DNA (covalently closed circular DNA (cccDNA) in the hepatocyte nucleus) continues to integrate into host DNA, contributing to hepatocarcinogenic mechanisms, as outlined above [30]. Future therapies to cure HBV have been proposed and are in early stages of clinical trials, including: inhibiting entry into hepatocytes and nucleus, interfering with transcription of mRNAs, capsidation, inhibition of reverse transcriptase (current NAs), augmenting the host immune system to fight and clear infected cells, and hopefully targeting cccDNA [30,34,47]. These provide hope for a future of less HBV-associated HCC cases by interrupting the ability of HBV to chronically infect a patient.

Patients with HBV associated HCC often present while asymptomatic but at an advanced stage, carrying a poor prognosis of weeks to months at best [18,38]. Certain factors have been associated with worse prognosis, such as higher HBV viral load, certain subgenotypes, local tumor microenvironment characteristics such as more T regulatory cells and cytokine profiles of T helper 2 cells versus T helper 1 cells, etc. [38,48]. This, similar to understanding the immune system’s role in the development of HCC, has led to advancements in treatment.

Treatment of HCC is patient-specific and dependent on how advanced the disease is, but largely consists of one or multiple therapies including: surgical resection, liver transplantation, chemo- or radiofrequency ablation, targeted chemotherapy, and systemic chemotherapies [18]. Some of the most significant recent advancements have been within the category of systemic options for advanced HCC. The introduction of sorafinib, a tyrosine and Raf kinase inhibitor, has improved survival [49]. Regorafenib, a multikinase inhibitor, and monoclonal antibodies nivolumab and pembrolizumab have been shown to help patients that have shown recurrence even after sorafinib [6,18,50,51]. Monoclonals in particular interact with the cell surface receptor PD-1, which is found on T regulatory cells [37,51]. As written by Roderburg et al., more research is needed to understand how biologic drugs can be targets for the innate immune system’s interaction with tumor [52]. Interestingly, the mechanisms of how chronic stress interacts with the body and can incite illness is similar to the mechanisms underlying these advancements in treatment that were in part informed by studies on tumor microenvironment.

## 3. Stress: Different Forms and Tools of Measurement

The term “stress” has become ambiguous as it is used loosely and often not clearly defined in the literature. The term was first coined by Hans Selye in 1973, who defined it as anything that caused deviation from a resting or balanced state in an organism, which was later defined as the concept of homeostasis [11,12,13,53,54]. Since its conception, the term is often used too broadly, leading to confusing the equity between two events that in reality are much different; not all stressors are created equal but may be labeled as so [53]. Kagan writes that the concept should be reserved for “events that seriously compromise the health or adaptiveness of select agents” [53]. As Somashekar et al. put it, stress is complex in that it is the combination of “the individual, environment, and interaction between them.” It results in a demand that evokes a response. Stressors—the thing or event that an individual responds to—can also be placed into categories including physiological, psychological, social, or environmental. The response to a stressor can also be categorized along with stressors, usually based on duration: acute, chronic, or episodic [11]. Chronic stress is usually thought to be greater than one month. Generally it is thought that acute responses are adaptive and can be healthy, whereas chronically they can have a negative influence on health. However, this may depend on the individual and the context they are in. As the American Psychological Association writes, physiologist John Mason and psychologist Richard Lazarus noted that the process in which an individual assesses a stressor is psychological but generally seen as threatening when demands outweigh resources at hand [55,56].

Although it is difficult to objectively measure stress, instruments have been created for use in the clinical setting. As outlined by Block et al., these include the General Health Questionnaire (GHQ-12), the Liver disease quality of live (LDQoL) instrument, Patient Health Questionnaire (PHQ-2 and 9), and the Perceived Stress Scale (PSS), among others [57]. Not only are these tools used in a practical clinical setting, but many studies that report on the effects of chronic stress and health have also used them as part of their research methodology.

As ambiguous as the term stress can be, there is still a multitude of examples in the literature on how it can health and health outcomes.

## 4. The Interaction between Chronic Stress, the Immune System, and Liver Disease

There are many proposed mechanisms in the literature that aim to explain the negative effects that stress can have on the body in general and more specifically, the liver. For example, psychosocial stressors have been shown to negatively impact health outcomes within cardiovascular disease, upper respiratory disease, HIV, autoimmune disease, and mental health, among others [14]. In general, a common theme is the interplay between different body systems influenced by the sympathetic nervous system in a physiologic stress response and its influence on deleterious outcomes [14]. For example, long-term sympathetic activation impacts the physiology of elevated blood pressure through its influences on vascular hypertrophy [58]. Further, interactions between the endocrine and immune system seem to play a more pertinent role within liver disease.

Joung et al. summarize that stress, through activation of different pathways, can overactivate one of the main immune cells in the liver, Kupffer cells [59]. Kupffer cells are residents in the liver and play an important role of maintaining baseline immune tolerance of the liver through secretion of anti-inflammatory cytokines [36,59]. Overactivation of these cells can lead to injury to the liver by activating other immune cells (including neutrophils) and production of reactive oxygen species. Joung et al. found that stress can incite overactivation of Kupffer cells through sympathetic nerve stimulation, alteration of hepatic blood flow, and intestinal flow of bacterial lipopolysaccharides [59]. These mechanisms might generally help explain why stress can negatively impact liver health and complicate liver disease further.

Many other studies have summarized stress as a risk factor for liver disease and a cause of increased morbidity and mortality. For example, Russ et al. found in a large meta-analysis in the UK that those with higher scores on a general health questionnaire that measures psychological distress also had a higher mortality from liver disease [60]. This was one of the first published studies that highlighted the significance of this risk factor. Other examples are more individual, including a study by Song et al. that found Japanese men with chronic perceived levels of stress had a higher incidence of liver cancer [16]. Vere et al. summarize research that has shown stress as a risk factor for viral hepatitis, cirrhosis, and hepatocellular carcinoma [61]. Kunkel et al. found that patients with depressive episodes and hepatitis B had elevated transaminases by displaying a correlation between laboratory values and Beck Depression Inventory [62]. In their research on patients with non-alcoholic steatohepatitis (NASH), Elwing et al. saw an association between major depressive disorder and generalized anxiety disorder and severity of NASH on histology [63]. Srivastava and Boyer found psychosocial stress to be a risk factor for relapse of type-1 autoimmune hepatitis [64]. These are just some of the many examples in the literature of how stress is linked to poorer outcomes among different diseases of the liver. The link with HBV-associated HCC will be explored next.

## 5. Chronic Stress as a Risk Factor for HBV-Associated HCC

As reviewed above, HCC may arise from chronic HBV infection through mechanisms that involve the immune system’s response to the virus. As commented on often in the literature, HCC is thought to be a very immunogenic cancer, which not only explains why viral infections like HBV start hepatocarcinogenic mechanisms but also how the tumor progresses [36,37,52]. For example, Roderburg et al. describe that the local tumor microenvironment, which highlights the initiation and response to tumor, is often composed of cells of the innate immune system [52]. They conclude that innate immune cells contribute to ongoing inflammation in the liver and incite hepatocarcinogenesis [52]. Granito et al. also explain the increased presence of T regulatory cells in the microenvironment and their role in suppressing the immune system’s response to the HBV virus and tumor itself [37]. Thus, the influence of chronic stress on HBV-associated HCC may be best understood by examining how stress and the immune system interact.

Chronic stress has been shown to influence the immune system not only in HCC formation, but also in HCC progression. As outlined above by Joung et al., stress can increase oxidative reactions that lead to local inflammation, causing liver cell damage that promotes mutagenic and carcinogenic mechanisms [59]. Other studies have looked at when HCC has already formed and how the microenvironment of the tumor itself changes to that of an immune suppresive profile. This would help the tumor continue to evade immune system attacks and may be why other studies have linked chronic stress to cancer morbidity and mortality [36]. For example, He et al. found that chronic stress causes cytokine profiles of the immune system to switch to those that are T helper 2 cell-mediated (higher IL-10 and lower IFN-gamma) and more immunosuppressive [65]. This is similar to the research that reported poorer prognosis in the setting of T helper 2 cell cytokine profiles, and also the studies that showed increased T regulatory cells that suppressed other pro-inflammatory immune cells [37,38]. Further, previous studies have shown that patients with chronic hepatitis B who were under chronic stress have lower levels of hair cortisol, a known stress hormone, displaying the downstream result of immunosuppression [66]. Other mouse studies observed that chronic stress causes changes in the immune cell makeup of the spleen to that of a less anti-tumor immune active profile and increased T regulatory cells [67,68].

The above discussion that focuses on the biochemistry and basic science mechanisms of HBV-associated HCC and stress (exemplified in Figure 1) are best translated to clinically relevant examples through case reports and review of family pedigrees. We reported a case of monozygotic twins who were both infected with HBV at birth but displayed extremely disparate outcomes [8]. One brother developed HCC, while the other twin remained a chronic HBV carrier. Since they shared the same virus genotype and similar genes, it can be theorized that non-genetic host factors such as life stressors could have contributed to poorer outcomes in one brother compared to the other. We also reported a case series of 33 patients with asymptomatic chronic hepatitis B followed over 32 years and highlighted the importance of the role of host through the differences in viral load fluctuations and ALT patterns over time [10]. In this series, many patients showed spikes in their viral load but had low levels of ALT, displaying the host–virus struggle. Many also had family members who developed complications of their disease, with likely the same genotype of virus. The fact that the individuals who remained asymptomatic for many years did not develop these complications may be in part due to non-genetic host factors such as chronic stress exposure or response to stress [10]. Another example is a story of a patient with cirrhosis secondary to chronic hepatitis B who was exposed to significant financial and emotional stress. At first, his routine screening MRI showed stable cirrhotic changes in the liver. Then, he experienced the stress of his long-term business closing and loss of insurance. MRI imaging two years later demonstrated a premalignant lesion (LI-RADS 4 lasting for 10 months). Several months later, he found a new job. Follow up MRI imaging showed that the LI-RADS 4 premalignant lesion had regressed to LI-RADS 3. Serial imaging thereafter continued to show regression to a benign form (LI-RADS 3 to LI-RADS 2) as his stressors continued to be relieved [69]. These examples highlight the influence of stress not only on the incidence of HCC in chronic hepatitis B but also on the progression (or regression) of tumor related to stress. Importantly, these examples of variation in the natural history of chronic hepatitis B included patients without other significant risk factors, such as alcohol use. Indeed, studies have confirmed that alcohol intake can increase the chance of developing HCV-associated HCC and mortality in HBV-infected individuals [70]. These examples also highlight that some individuals are likely more prone to stress and the effects that stress has on the body.

Both Schneiderman et al. and Yaribeygi et al., in their review of stress on health, describe how people with better coping mechanisms may not see the same deleterious effects of stress as those that do not cope well [14,15]. For example, Nagano et al. found an association between personalities that have higher levels of chronic stress and elevated transaminases in the setting of Hepatitis C [71]. Thus, not only might those with stress in their lives have higher morbidity and mortality from disease compared to those without, it may also matter how the individual responds to stress.

Bonkovsky also challenges the findings of Russ et al. and raises an important question in the link between stress and liver disease [72]. He writes that it may be difficult to assess whether stress itself is a risk factor for developing liver disease or if having liver disease is what leads to more stress. However, Russ et al. attempted to control for this in their study and found it likely not to be the case [72].

These points raise the question of who is at higher risk of stress-related harmful effects on health and who is at risk of becoming stressed to begin with. Indeed, one of the most profound sources of stress is that of a psychosocial stress. Racial disparities have been found to be one such source.

## 6. Racial Disparities as a Subset of Chronic Stress and HBV-Associated HCC

Although some individuals may be more prone to stress and cope with stress differently, others are less fortunate to have direct control over whether they experience stress or not. Psychosocial stress in the form of finances, for example, may be a result of uncontrollable disparities that are structural and systems-based in America. The WHO classifies stress as one of the top 10 social determinant of health inequities [73]. As Brondolo et al. write, “individuals facing economic and social disadvantage experience more subjective and objective threats but often have fewer resources to respond to these threats” [55]. This expands on the basic definition of stress occurring when demand outweighs resources. Minorities in America are thus not only more likely to experience socioeconomic stressors but be less equipped to handle them. They add that multiple stressors can also start to cluster from different sources, and that those with social and economic disadvantages are more likely to accumulate these. Racial disparities are an example of this, especially within healthcare.

Health-related racial disparities in America continue to be an ongoing problem. These disparities may exist for many reasons, including the different exposure to stressors that peoples of different race have. For example, Sternthal et al. found in their study that Blacks had the highest prevalence of stress compared to Whites, with financial and relationship stressors being the most impactful on poorer health outcomes [74]. Further, the more an individual is exposed to stress, the more likely they may be to practice behaviors that are not health-promoting (such as eating unhealthy foods, lack of physical exercise, etc.) [55].

HCC outcomes are not immune to racial disparities in America. Indeed, research shows that African-Americans have higher incidence of HCC in America but are less likely to receive surgery, more likely to wait longer for surgery if they have it, and have worse outcomes overall [75,76,77,78,79]. These disparities are undoubtedly a source of stress on an individual. Chayanupatkul et al., in their study examining populations with and without cirrhosis, found that more African-Americans in the group without cirrhosis developed HCC compared to the group with cirrhosis [80]. They did not find a specific factor that contributed to this difference seen. However, it may be postulated that this risk is in part increased by the deleterious effects of stress on health and the risk of HCC in the form of psychosocial racial disparities in America. Further research may help us better understand what specific interventions or support provided to those at highest risk can help mitigate the progression to developing HCC. Indeed, recognizing the data that supports minorities have worse outcomes is certainly a place to start.

## 7. Conclusions

HCC ranks as one of the most common cancers in America and globally. HBV is an important and relevant risk factor for the development of cirrhosis and HCC. Understanding how the Hepatitis B virus replicates in the cell better explains how it can contribute to direct viral hepatocarcinogenic mechanisms and interact with the immune system to do the same. The vagaries of the host responses to virus and resultant long-term outcomes such as which individuals develop HCC remain a topic of perplexity. We highlight one important risk factor often overlooked in the medical community: stress. Though typically vaguely defined, we explore that significant demands in life that outweigh available resources, especially in the chronic setting, can have deleterious effects on health. Stress has not only been found to have adverse effects on multiple organ systems in the body in previous research, but has also been linked to increased cancer incidence, morbidity and mortality. The complex interaction between stress and the immune system in particular may explain why it is an important risk factor for HBV-associated HCC. This cancer is known to be an immunogenic tumor due to the local tumor microenvironment interaction with the innate immune system. Stress and the resultant physiologic response not only allows carcinogenesis by instigating inflammation, but also allows a tumor to progress through immune suppression and dysregulation. Some individuals are notably better at coping with stress and may be at a lower risk for developing HBV-associated HCC. However, some populations are more vulnerable to stress, including minority races that experience disparities in healthcare and outcomes in America. Further research is needed to better describe the intricacies of HBV-associated HCC and stress, but this review highlights the need for clinicians to recognize it as an important and potentially preventable risk factor.

## Figures and Tables

**Figure 1 ijms-23-03917-f001:**
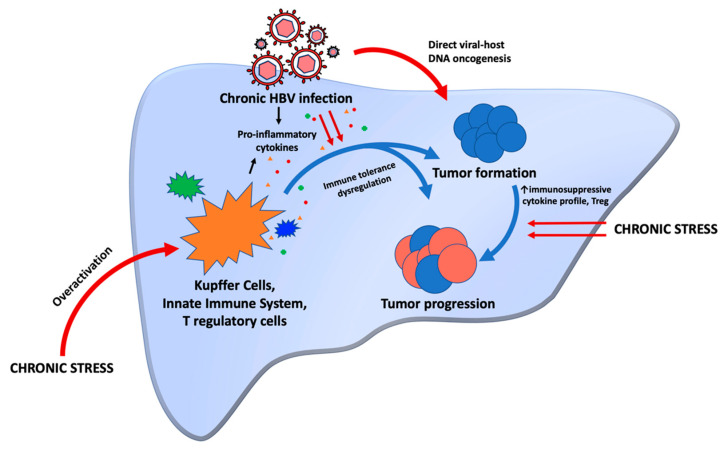
Chronic stress can lead to immune system overactivation, which in combination with chronic HBV infection, leads to immune tolerance dysregulation. Pro-inflammatory cytokines released by immune cells in response to chronic stress and chronic HBV infection also contribute to tolerance dysregulation. This helps tumors to form. HBV DNA also directly integrates into host DNA, another oncogenic mechanism in the formation of tumors. Once a tumor is formed, chronic stress can also switch cytokine profiles to more immunosuppressive ones that allow for the tumor to progress.

## Data Availability

Not applicable.

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
