# Peer review of "Hepatitis B Virus-Associated Hepatocellular Carcinoma and Chronic Stress"

_ijms, 2022, doi:10.3390/ijms23073917_

Round 1

Reviewer 1 Report

This is an interesting review discussing Hepatitis B virus related hepatocarcinogenesis, focusing on the impact of stress in tumor initiation and progression. 

-The authors should include how HCC epidemiology is changed in the last decades with an increasing role of metabolic etiology and improvement of antiviral therapies for HBV and HCV chronic liver diseases, as recenlty reported (The evolutionary scenario of hepatocellular carcinoma in Italy: an update. Liver Int. 2017 Feb;37(2):259-270).

-HBV-related HCC: the authors should recall the prognostic impact of HBV etiology in HCC outcome and response to systemic treatment and in particular to regorafenib systemic therapy as recently reported ( Identification of Regorafenib Prognostic Index (REP Index) via Recursive Partitioning Analysis in Patients with Advanced Hepatocellular Carcinoma Receiving Systemic Treatment: A Real-World Multi-Institutional Experience. Target Oncol. 2021 Sep;16(5):653-661; Experience with regorafenib in the treatment of hepatocellular carcinoma. Therap Adv Gastroenterol. 2021 May 28;14:17562848211016959. )

Author Response

  • Thank you for this suggestion regarding epidemiology. This has been further clarified in the introduction, on lines 30-32, with more citations added as corresponding evidence. 
  • Thank you for calling attention to this. Discussion of prognosis and treatment has been added in lines 156-176, as well as the citations above and others relevant to this information. 

Reviewer 2 Report

This is a well-written review article on Hepatitis B virus related hepatocarcinogenesis. Among involved factors, they focused on the impact of stress on the body as it might potentiate these mechanisms. In particular, they discuss how in hepatocellular carcinoma, this can lead to tumor initiation and progression. 

The manuscript is of current significant interest and well presented. However, it could be improved the discussion of some points that deserve further data and should be addressed.

-HBV-related HCC: Proposed mechanisms: regarding HCC risk in HBV chronic liver disease, the authors should recall recent literature data on the pathogenetic role of the CD4+ CD25+ regulatory T cells (Treg) in the HCC development as recently reported (Hepatocellular carcinoma in viral and autoimmune liver diseases: Role of CD4+ CD25+ Foxp3+ regulatory T cells in the immune microenvironment. World J Gastroenterol. 2021 Jun 14;27(22):2994-3009). This may be of relevance discussing The interaction between chronic stress, the immune system, and liver.

-The authors should also recall the potential impact of alcohol intake in the risk of HCC development in HBV-related chronic liver disease as previously demonstrated (Natural course of chronic HCV and HBV infection and role of alcohol in the general population: the Dionysos Study. Am J Gastroenterol. 2008 Sep;103(9):2248-53).

Author Response

  • Thank you for these comments. Pathogenesis of HCC development and progression related to T regulatory cells has been added to the manuscript to both the sections HBV-related HCC and chronic stress as a risk factor for HBV-related HCC along with relevant citations. Figure 1 was also edited to include more details regarding T regulatory cells. 
  • Thank you for this suggestion. This has been added to the discussion on chronic stress as a risk factor for HBV-related HCC on lines 308-312.